# Does Immunocastration Affect Behaviour and Body Lesions in Heavy Pigs?

**DOI:** 10.3390/vetsci9080410

**Published:** 2022-08-05

**Authors:** Gaia Pesenti Rossi, Emanuela Dalla Costa, Joel Fernando Soares Filipe, Silvia Michela Mazzola, Ambra Motta, Marzia Borciani, Alessandro Gastaldo, Elisabetta Canali, Federica Pilia, Marco Argenton, Mario Caniatti, Alessandro Pecile, Michela Minero, Sara Barbieri

**Affiliations:** 1Department of Veterinary Medicine and Animal Science, Università degli Studi di Milano, 26900 Lodi, Italy; 2Fondazione C.R.P.A. Studi Ricerche, 42121 Reggio Emilia, Italy

**Keywords:** heavy pig, immunocastration, animal welfare, behaviour, body lesion, salivary testosterone, productive performances, carcass traits

## Abstract

**Simple Summary:**

Castration of piglets is routinely practiced to prevent the presence of boar taint. In Europe, around 61% of pigs are surgically castrated, 54% of which do not receive any pain-relief therapy. The increasing public awareness towards animal welfare and the ethical characteristics of animal-derived products has led to the necessity of finding more animal-friendly alternatives. Among them, immunocastration, a process of active immunisation leading to the suppression of testicular function, has shown promising results in light pig production, both in terms of animal welfare and productivity. Besides, no study has yet investigated animal welfare in immunocastrated males intended for Italian heavy pig production. This study aimed at evaluating the effect of immunocastration on the welfare of heavy pigs by monitoring behaviour and body lesions in immunocastrated and surgically castrated pigs. Salivary testosterone levels and productive traits were also evaluated. Our results confirmed that immunocastration is a suitable alternative to surgical castration with profitable productive performance, whereas the higher activity of animals and the higher presence of body lesions before the effective immunisation caused an impairment of animal welfare, which should be further investigated as a critical aspect in heavy pig production.

**Abstract:**

Immunocastration has been pointed out as an alternative to surgical castration; though, most of the scientific studies were performed in light pig production. This study aimed to evaluate the effect of immunocastration on animal welfare in heavy pig production through the evaluation of behaviour and body lesions. A total of 188 commercial-hybrid pigs were randomly allocated into two treatment groups: surgical castration (SC) and immunocastration with Improvac^®^ (IC). Data on behaviour, body lesions, and salivary testosterone levels were collected the day before each vaccination at 15, 22, 32, and 36 weeks of age. IC and SC pigs were slaughtered at 40 and 41 weeks of age, respectively; productive and carcass traits data were also collected. Considering productive performance, our results confirmed that IC pigs grew faster and presented a higher weight at slaughter. A critical period for pig welfare was observed before 32 weeks: testosterone concentration and body lesion score were significantly higher in IC pigs compared to SC pigs; active behaviours were significantly more frequent in IC at 15 weeks. Immunocastration may represent a suitable alternative to surgical castration with profitable productive performances, whereas the impairment of welfare during the period before the effective vaccination should be further investigated as a potential critical aspect in heavy pig production.

## 1. Introduction

Around 61% of European pigs are surgically castrated, 54% of which do not receive any pain-relief therapy [1,2], as permitted in Europe by Directive 2008/120/EC [3] for piglets younger than seven days of age. Castration of male piglets has been traditionally practised to prevent the presence of boar taint, a sex-specific off-odour caused by two substances, androstenone and skatole [4,5,6,7], which may be found in boars. However, it has been widely demonstrated that castration without anaesthesia or analgesia impairs animal welfare, causing pain, stress, and discomfort [8,9]. Immunocastration has shown promising results as a possible alternative to surgical castration: it consists of active immunisation against the gonadotropin-releasing hormone (GnRH), followed by the suppression of the hypothalamic-pituitary-gonadal axis and the suppression of testicular function [10,11]. According to manufacturer instructions, several studies have evaluated the effect of a two-dose protocol on pigs reared for light pig production, with animals slaughtered at a mean age of 24 weeks and around 120 kg live weight [12,13,14,15]. To date, only one study evaluated the use of immunocastration in male pigs in the Italian heavy pig production system, where animals are slaughtered at a minimum of 36 weeks of age and around 180 kg live weight and suggested the necessity of a third dose [16].

Immunocastration is reported as a more animal welfare-friendly alternative not only in pigs but also in cattle and small ruminant husbandry [13,17,18,19,20]. It prevents distress and pain caused by surgical castration and its correlated risks, overcoming the higher mortality present in surgically castrated pigs in the first week post-partum [21]. The procedure usually causes only mild reactions at the injection site [15], and the associated pain is usually limited only to the needle insertion. After the second injection, vaccinated pigs show less aggressive and sexual behaviour than boars [22,23,24]. However, until then, immunocastrated pigs are physiologically entire males and display male-like behaviour, showing more aggressive and mounting episodes, resulting in a higher number of skin lesions [14,24]. Schmidt et al. [25] reported a higher incidence of skin lesions caused by mounting behaviour in immunocastrated pigs before the effective dose, which disappeared after the second injection. These findings may raise concerns regarding the adverse impact of the technique on the welfare of pigs.

Regarding productive performances, published data reports that immunocastrated pigs tend to grow faster and have a better feed conversion ratio than surgically castrated males, thanks to both the exploitation of the full growth potential of entire males and the benefits of castrate-like metabolism [13]. Immunocastration also leads to a heavier carcass, a higher percentage of lean meat, and lower fat thickness [26,27].

The vast majority of studies focusing on behavioural, welfare, or productive aspects of immunocastrated pigs concern light pig production, while currently, no study has yet investigated animal welfare in immunocastrated pigs throughout the fattening period in heavy pig production.

This study aimed to compare the behaviour and body lesions of immunocastrated and surgically castrated heavy male pigs from growing to slaughter. Salivary testosterone levels and productive traits were also considered.

## 2. Materials and Methods

The study was approved by the Animal Welfare Committee of the University of Milan (OPBA_26_2020) according to Directive 2010/63/EU. A total of 188 commercial-hybrid male pigs (Topigs Norsvin, Helvoirt, The Netherlands) were enrolled in this study. The research was conducted on two different intensive farms for the production of heavy pigs in the North of Italy. Animals were housed and managed in compliance with Council Directive 120/2008/EC. Buildings were naturally ventilated and provided with a minimum of 8 h of artificial light. Diets were formulated to meet the nutritional requirements of the growing-fattening phase, according to the National Research Council Nutrient Requirements of Swine [28]. Both in the growing and in the fattening periods, pigs were not mixed and were housed under similar housing and feeding conditions. Pigs were monitored daily for their growth and general health condition by the farmer. To assure that all the animals within the experimental groups maintained a good health standard at the moment of the vaccination, pigs identified through the rearing process as “problematic” from a health point of view or showing an inadequate body weight were then separated and managed according to their conditions.

During the growing phase (35 kg to 75 kg live weight), pigs were housed separately in 2 pens with straw bedding (1 m^2^/pig for a BW of 70 kg). Each pen was provided with a self-feeding system and eight nipples. Animals were fed ad libitum with a dry commercial diet. In the fattening accommodation, pigs (75 kg to slaughter) were housed separately in 8 different pens with a fully slatted concrete floor (1.06 m^2^/pig for BW of 160 kg) and fed with a liquid commercial diet in a trough three times a day. Water was provided ad libitum by two nipple drinkers in each pen. Metal chains with wooden bars hung on the walls were provided in each pen as enrichment material. 

At the end of the rearing period (at 40–41 weeks of age), pigs were slaughtered according to the routine abattoir process. 

### 2.1. Treatments

At birth, male piglets were randomly allocated to 2 treatment groups: surgical castration (SC) and immunocastration (IC). Animals of the SC group (*n* = 94) underwent surgical castration at four days of life, according to Council Directive 2008/120/EC and standard farming procedures. The immunocastration group (*n* = 94) underwent vaccination with Improvac^®^ (Zoetis Italia Srl, Roma, Italy). The vaccine was injected subcutaneously with a 2 mL vaccine gun by a qualified veterinarian immediately behind the ear. In the Improvac^®^ package leaflet, it is suggested to perform the first administration from 8 weeks of age and the second 4 to 6 weeks before slaughter; it is also suggested that, if slaughter is later than 10 weeks after the second dose, a third dose should be given 4 to 6 weeks before the planned slaughter date. According to these indications, the local supplier proposed a 3-dose regimen on the basis of a recent effective treatment on the same farm with animals of the same breed. The protocol proposed included the administration of Improvac^®^ at 15 weeks, 24 weeks of age, and 5 weeks before slaughter (36 weeks of age).

However, due to a high level of aggression in the IC group, the second vaccination was anticipated and then repeated after two weeks, according to supplier suggestions; a fourth intervention was added. More details on the timing of vaccination are reported in Table 1. After each Improvac^®^ administration animals were monitored for possible adverse reactions, namely the presence of injection site swellings, which is reported in the leaflet as commonly observed, or the presence of anaphylactoid-type reactions, which is reported to happen in very rare cases.

### 2.2. Data Recordings

Data were collected the day before Improvac^®^ injections when animals could display the highest behavioural expression, as vaccine effect is expected to be the minimum. According to this assumption, four timepoints (TP) were established: T0, T1, T2, and T3, respectively, at 15, 22, 32, and 36 weeks of age. Data collected from each timepoint was used to investigate any treatment effects.

#### 2.2.1. Behavioural Observations

Video recordings of pig behaviour were carried out with a stationary camera (GoPro Hero 7) fixed to a pole at the corner of each pen about 2 m above ground, without interfering with normal pig behaviour. At each timepoint (T0, T1, T2, and T3), animals were recorded for 45 min between 9:30 am and 11:30 am (before food administration). An instantaneous scan sampling method was used to score pigs’ behaviour [29,30]. A 30 s sample interval was used, leading to a total of 90 scans for each TP. Observed behavioural categories and their related descriptions are reported in Table 2. The sampling procedure aimed to obtain an estimate of the proportion (or percentage) of individuals in each behavioural category presented.

#### 2.2.2. Body Lesions

To assess animal agonistic behaviour, body lesions were evaluated at each TP based on a validated scoring system [31]. Each pig was visually assessed by one trained observer by inspecting one side of the pig’s body, considering five anatomical regions: ears, front, middle, hindquarters, and legs. Each considered region was scored as 0, 1, or 2 depending on the number of lesions. Injuries that could be attributed to the pen structures were not considered. Finally, the body lesion score of each animal was calculated by summing the single scores of each region. Tail biting was also assessed, separately, according to the Welfare Quality Assessment [31]. 

#### 2.2.3. Testosterone Levels

Saliva samples were collected at each TP to quantify testosterone levels in the 2 treatment groups. Considering the different group sizes in the two housing conditions and to reduce the sampled pigs, a different number of animals was selected for testosterone analysis during the experimental period: for saliva collection, 25 pigs for each group were randomly selected at T0 and T1, while 12 pigs (3 subjects × 4 pens) for each group were randomly selected at T2 and T3. To minimise the possible stress related to the procedure, saliva was collected from pigs that voluntarily approached the operator. Unrestrained pigs were then allowed to chew a cotton swab (Salivette, Aktiengesellschaft & Co., Sarstedt, Nümbrecht, Germany) held by long forceps for at least 1 min until thoroughly moist. Salivette^®^ rolls were then placed into centrifugation tubes and kept at 4 °C. After centrifugation at 2000× *g* for 5 min, supernatants were retrieved and all samples were analysed using commercially available ELISA kits (Salimetrics LLC, State College, PA, USA), according to the manufacturer’s instructions. The samples were stored at −20 °C until the day of analysis. The sensitivity and detection range of the assays is 0.1–20 ng/mL. Briefly, testosterone present in the samples competes with testosterone conjugated to horseradish peroxidase for the antibody binding sites on a microtiter plate. The optical density is read on a standard plate reader at 450 nm. The amount of testosterone enzyme conjugate detected is inversely proportional to the amount of testosterone present in the sample.

### 2.3. Growth Performance and Carcass Traits

Animals were weighed when relocated to growing and fattening accommodation and before moving to the slaughterhouse. An average daily gain (ADG) was calculated. Carcasses were weighed and classified according to Decision 38/2014/EC using the Fat-O-Meter system, which measures backfat (F) and longissimus dorsi muscle (M) thickness between the third- and the fourth-last rib, 8 cm off the midline of the split carcasses. Lean meat content was expressed as a percentage (%) and calculated by combining the two measurements using the equation:LM (%) = 45.371951 − 0.221432 × F + 0.055939 × M + 2.554674 × (M/F)

Based on lean meat content, carcasses were then classified using the European SEUROP system, according to Reg 1308/2013/EC and Reg 1182/2017/EC. 

### 2.4. Statistical Analysis

Statistical analysis was performed using SPSS 27 (SPSS Inc., Chicago, IL, USA). Statistical significance was accepted at *p* ≤ 0.05. The data were tested for normality and homogeneity of variance using the Kolmogorov–Smirnov and Levene test, respectively. The Mann–Whitney U test was used to investigate possible differences between treatment groups at each timepoint for the considered variables (percentage of active and inactive pigs, body lesion score, testosterone level, and carcass traits).

## 3. Results

None of the animals in the present study showed adverse reactions to Improvac^®^ administration.

A total of 26 SC pigs and 23 IC pigs were excluded from the study because they did not reach production standards or showed poor health conditions during the trial. Eleven surgically castrated pigs and 11 immunocastrated pigs were excluded during the growing phase, and the same number of subjects for each group were excluded during the fattening phase. The excluded animals did not cause any numerical disproportion in the groups. At the time of loading, 4 SC pigs and 1 IC pig were not sent to the slaughterhouse due to inadequate body weight. The outcome of the excluded pigs was not known, but it was not linked to vaccine administration. 

Table 3 summarises group size. One surgically castrated pig was excluded from carcass processing because of slaughterhouse technical problems.

### 3.1. Behavioural Observations

The percentage of active pigs is presented in Figure 1. Compared to IC, SC pigs were significantly more active at T0 (Mann–Whitney U test, *p* = 0.000). The activity level of IC pigs increased and was significantly higher compared to SC pigs at T1 and T2 (Mann–Whitney U test; *p* = 0.000 and *p* = 0.001, respectively). No statistical differences between treatment groups were found at T3.

The proportion of inactive pigs followed an opposite trend: IC pigs rested significantly more at T0 compared to SC (Mann–Whitney U test, *p* = 0.000); while at T1 and T2, SC were more inactive compared to IC (Mann–Whitney U test; *p* = 0.000 and *p* = 0.001, respectively). No statistical differences between treatment groups were found at T3.

### 3.2. Body Lesions

Results of body lesion score (mean ± 1SE) are presented in Figure 2. At T0, before the first administration of Improvac^®^ (15 weeks of age), IC pigs showed a significantly higher body lesion score (0.60 ± 0.11) compared to SC (0.2 ± 0.05) (Mann–Whitney U test, *p* = 0.004). A significant difference was also found between treatments at T1 (22 weeks of age), with IC pigs showing significantly higher body lesion scores (0.33 ± 0.06) compared to SC (0.19 ± 0.05) (Mann–Whitney U test, *p* = 0.039). At T2 and T3, no significant difference in body lesion score was found.

Ear and leg lesions were not considered for further analysis due to confounding factors (i.e., the high presence of ear necrosis and the deep litter in the growing phase, respectively). Eight pigs over the entire experimental period, equally divided into the two groups and at different timepoints, presented tail lesions caused by tail biting behaviour.

### 3.3. Testosterone Levels

In order to evaluate testosterone concentration, an adequate quantity of saliva was easily collected without causing stress to the animals. The testosterone levels (mean ± 1SE) are presented in Figure 3. Compared to SC, IC pigs had a significantly higher testosterone level in T0, T1, and T2 (Mann–Whitney U test; T0 *p* = 0.000; T1 *p* = 0.000; T2 *p* = 0.000). T3 testosterone concentrations were similar in IC and SC pigs (*p* = 0.204). 

### 3.4. Growth Performance and Carcass Traits

IC and SC pigs showed similar weights at the beginning of the growing (IC = 35.22 kg; SC = 35.77 kg) and fattening phases (IC = 77.95 kg; SC = 77.12 kg). In the growing phase, the average daily gain was 690 g in IC and 670 g in SC pigs, while in the fattening phase it was 820 g in IC and 710 g in SC pigs. Despite the one-week shorter fattening period, IC pigs were significantly heavier at the end of the rearing process (Mann–Whitney U test; *p* = 0.002), with a mean weight of 180.99 ± 14.54 kg, while SC pigs weighted 171.32 ± 12.52 kg.

Carcass traits results are reported in Table 4. Hot carcass weight resulted significantly higher (Mann–Whitney U test; *p* = 0.002) for IC pigs (150.54 ± 1.48 kg) compared to SC pigs (145.10 ± 1.31 kg). The mean fat (30.38 ± 0.59 mm) and muscle thickness (55.34 ± 8.94 mm) of IC resulted in a higher mean lean meat content (51.67%). Conversely, SC pigs showed a higher mean value of fat (32.31 ± 4.72 mm) and muscle thickness (58.47 ± 6.48 mm), resulting in a lower lean meat content (50.86%). The fat and muscle thickness were significantly different between groups (Mann–Whitney U test; *p* = 0.034 and *p* = 0.028, respectively). IC pig carcasses were classified as follows: HU = 50; HR = 18; HE = 3; SC pig carcasses: HU = 42; HR = 24; HE = 1. The four “E” carcasses (3 IC, 1 SC) were excluded from the Parma Circuit. 

## 4. Discussion

The present study focused on the effect of immunocastration on the welfare of pigs in heavy pig production, considering the impact on behaviour and body lesions. 

Our results confirmed that the preservation of pig welfare is achieved after the effectiveness of immunocastration action (second dose). 

The immunocastration protocol included in the present study consisted of three doses that should have been administered respectively at 15, 24, and 36 weeks of age. The initial protocol was changed during the study to maintain animal welfare at an acceptable level; the second vaccination was anticipated (22 weeks of age) and then repeated after two weeks (24 weeks of age) due to the high level of agonistic behaviour. Thereafter, a subsequent intervention was added as suggested by the local supplier (32 weeks of age). The last administration was given five weeks before slaughter (at 36 weeks of age). Therefore, animals received a total of five doses; this greatly differs from what has been reported in the literature, where a three dose protocol was proposed for Italian heavy pig production, with interventions at 10–11, 26–27, and 36–37 weeks of age [16]. A three dose protocol is also used for male pigs with long life cycles like Iberian pigs. Pérez-Ciria et al. [32] assessed the effect of immunocastration on male and female pigs and three types of experimental diets on meat and fat quality for Teruel dry cured ham. In this study, male pigs received the three doses at around 8, 14, and 17 weeks of age and were slaughtered between 25 and 28 weeks. Font-i-Furnols et al. [33] also evaluated ham traits and meat quality in immunocastrated pigs injected with three doses of Improvac^®^ at 4.5, 5.5, and 9 months of age or 11, 12, and 14 months of age and slaughtered at 17 months. However, these studies focused on productive performance and carcass quality traits, underlining the technique’s capability to eliminate boar taint effectively, but no information was provided about welfare issues. Immunocastration protocols were also evaluated in heavy female pigs. In a study about Iberian female pigs [34] it was seen that both a 3- and a 4-dose Improvac^®^ regimens were efficacious in suppressing oestrus, reducing the incidence of standing oestrus, serum progesterone levels, and the development of the uterus and ovaries. Therefore, Dalmau et al. [34] stated that a 3-dose regimen was sufficient for animals up to 14 months of age to suppress oestrus and its related behaviour. However, also in this case, no welfare issues (nor behavioural recordings other than standing oestrus, or skin lesions) were specifically reported. These aspects were evaluated in a study by Di Martino et al. [35], in which female heavy pigs received three doses of Improvac^®^ at 16, 20, and 32 weeks of age (4 weeks before slaughter), with no issues related to the protocol used being assessed. To the best of our knowledge, no previous study observed such a high level of aggressive interactions as to modify the initial protocol, anticipate the vaccination, and increase the number of administrations. We hypothesised that the second dose, injected at 22 weeks of age, failed for some reason to activate the immune response and did not suppress aggressive behaviour; therefore, it was readministered two weeks later (V2.2). Moreover, V3 (administrated at 32 weeks of age) was added according to the local supplier, as animals previously did not seem to respond correctly to the vaccination; this condition was subsequently confirmed by the high level of testosterone found just before the administration. These two aspects, namely the necessity to anticipate the administration and the failure to suppress testicular function, might be linked on one hand to the genetics of the animals and on the other hand to the farm management. In the literature, the presence of animals responding poorly to the vaccinations, the so-called “non-respondents”, is reported in around 0–3% of vaccinated animals and is related to either accidentally missed vaccinations or to health problems in animals during the immunisation [13,18]. This percentage, however, does not correspond with our findings, as the vast majority of animals showed aggressive behaviour and a high testosterone level. Further study should explore the possible causes of non-respondent issues in heavy pig production. 

Our results show that the most critical period for the welfare of immunocastrated heavy pigs is before the effective Improvac^®^ administration (received at 24 weeks of age). Compared to surgically castrated pigs, IC pigs were more active (e.g., explorative and agonistic behaviour) and had a significantly higher body lesion score (probably linked to agonistic and sexual behaviour, such as mounting). Both the activity and the higher presence of body lesions in IC pigs could be related to the fact that, before the second administration of Improvac^®^, pigs were physiologically similar to entire males. Studies have underlined how entire males and immunocastrated pigs before the second vaccination display more activity and, in particular, a higher frequency of aggressive and mounting behaviour than surgically castrated males [23]. It is also reported that entire males present more skin lesions when compared to surgically castrated pigs or pigs after full immunisation [14,36,37]. A higher presence of skin lesions before the effective dose of Improvac^®^ was also found by Schmidt et al. [25], especially in the shoulder region, caused by mounting behaviour; subsequently, no difference in lesion score was measurable between the groups after the second vaccination. Both behavioural aspects and body lesion scores improved after the effective dose. Although the percentage of active pigs significantly differed between IC and SC pigs at 32 weeks of age, it was lower if compared to the previous timepoints. At 36 weeks of age, no significant difference was detected, and the percentage of both IC and SC active pigs was the lowest recorded during the experimental period. At 32 and 36 weeks of age, IC and SC pigs showed similar values for body lesion scores. These findings are in accordance with the literature; after the effective dose, immunocastrated pigs are quieter, and fewer agonistic or sexual behaviours are performed, with a lower presence of skin lesions, resembling more barrows than boars [14,24,25,37,38]. Overall, the general reduction of activity in both groups during the rearing period was also observed in the studies of Brewster and Nevel [38] and Di Martino et al. [35] in female heavy pigs. Our findings have shown an increase in resting behaviour at 32 and 36 weeks of age, independently of the treatment; this is in accordance with the literature, where a marked increase in resting behaviour (up to 70–80%) at the end of the fattening period has been reported [35,39,40]. Salivary testosterone was sampled just before the subsequent administration to evaluate the level achieved at each time point by animals from different groups. An adequate quantity of saliva was collected in a non-invasive way and the analysis provided reliable results without causing any stress to the animals, confirming this technique as a simple, feasible, non-invasive method to evaluate testicular function. Compared to the SC group, immunocastrated pigs had a significantly higher testosterone concentration at all the considered timepoints except for the last, gradually increasing from 15 weeks of age (before the first administration) and peaking at 32 weeks of age, seven weeks after the effective dose, and then dropping at 36 weeks of age to levels lower than SC pigs. Assuming V2.2 (24 weeks of age) was the effective dose, the high level of testosterone found 7 weeks after contrasted with what is reported in the literature: Claus et al. [41] observed that testicular function, and consequently the release of hormones and androstenone, was safely blocked from the eighth day after the effective dose administration and remained inhibited for 10–24 weeks, depending on the subject. Another study reported that the reduction in testosterone lasted up to 22 weeks after the second administration [11]. However, as mentioned before, although at 32 weeks of age IC pigs showed high testosterone concentration, their body lesion score was not significantly different from that of SC pigs and was lower if compared to the previous timepoints. Moreover, the reduction of activity found at this timepoint is in accordance with the literature. Many studies showed that before the second injection, the activity of vaccinated pigs was as high as that of entire male pigs, whereas, after the second injection, the activity decreased to a similar level as that of surgically castrated pigs [14,22,24,37]. 

Finally, IC pigs grew faster, maintaining a higher average daily gain than SC pigs, resulting in a higher weight at slaughter, despite the one week shorter fattening period. IC pig carcasses also showed lower fat and muscle thickness, resulting in a higher lean meat percentage. Our results are in accordance with the sole study dealing with the use of immunocastration in Italian heavy pig production [16], confirming that this technique is an interesting alternative to surgical castration in this production system, as neither productive performance nor quality were negatively influenced. Immunocastrated heavy pigs showed good productive performance, confirming widely demonstrated aspects such as faster growth, higher weights at the end of the rearing process, and leaner carcasses [26,27]. These aspects have been correlated in the literature to the anabolic advantage during the period before the second dose, when animals are yet formally entire males. On the other hand, some studies have also focused on the changes in feeding behaviour; interestingly, it has been reported that feed intake behaviour increases by 25% in immunocastrated pigs [42]. However, an increased feed intake and related fast growth might not be desirable in heavy pigs. In this production system minimum age and weight at slaughter are requested. An appropriate feeding regime must be defined, considering that a restriction or a dilution of the diet can further exacerbate aggressive behaviours and increase the incidence of skin lesions, resulting in further welfare issues [18]. In addition, the necessity of a higher number of Improvac^®^ doses raised concerns around the economic sustainability of immunocastration in heavy pig production. In light pig production, it was estimated a cost of EUR 3–3.65 per pig, considering both vaccination and labour costs [43]. It should be further evaluated if productive advantages reported can overcome the higher cost of the protocol. Another point worth considering is the aspect of the commercialisation of the meat from immunocastrated pigs. It deals not only with the acceptability of it by the consumers, which has been positively expressed in a cross-country study by Vanhonacker and Verbeke [44], but also with the perception of risk tied to the application of immunocastration by consumers and the relative willingness to accept this technique. These aspects were recently evaluated in Italian consumers by Di Pasquale et al. [45], who found out that there is a perceived health-related risk tied to the application of immunocastration, but the perception of this risk has different levels within the population. Moreover, immunocastration was the preferred alternative technique for more than 30% of the sample. These aspects highlight that immunocastration is perceived as a welfare-friendly alternative to surgical castration, but more information must be provided to the consumer in order to accept this technique with a more positive attitude.

## 5. Conclusions

The results of the present study suggest that immunocastration positively affects animal welfare after the effective Improvac^®^ vaccination. However, a critical impairment in pig welfare was observed before the full immunisation, as confirmed by the higher activity of animals and the higher presence of body lesions. From a productive point of view, immunocastration in heavy pigs is an advantageous alternative to surgical castration. Our results also suggest that at least four administrations of Improvac^®^ are required to reduce testosterone levels, control aggressive behaviour, and reduce body lesion score. Hence, this raises concerns about the economic sustainability of this technique when applied to the heavy pig production system. Additionally, monitoring salivary testosterone results in a useful on-farm tool to define the onset of puberty in animals, as saliva collection is non-invasive and well tolerated by the pigs.

Further studies should focus specifically on immunocastrated heavy pigs before the effective vaccination; better data comprehension might be achieved through the evaluation of testosterone levels at regular intervals and monitoring specifically the aggressive and sexual behaviour trends. However, the best immunocastration protocol to use in heavy pig production has yet to be identified in order to ensure an adequate level of animal welfare throughout the entire rearing process, while also maintaining acceptable productive traits.

## Figures and Tables

**Figure 1 vetsci-09-00410-f001:**
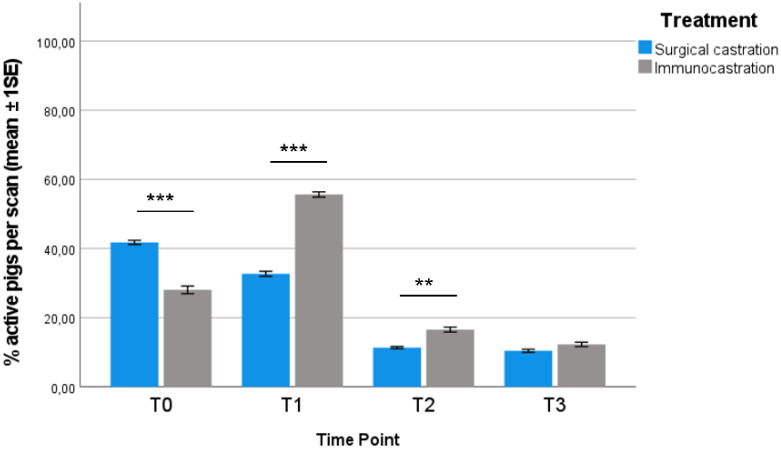
Mean percentage (±1SE) of active pigs (defined as animals busy in one of the following: walking, exploratory behaviour, social and agonistic interactions, sniffing, biting, chewing, or exploring environmental enrichment) per scan in each TP for each treatment group considered (Mann–Whitney U test; ** *p* < 0.01; *** *p* < 0.001).

**Figure 2 vetsci-09-00410-f002:**
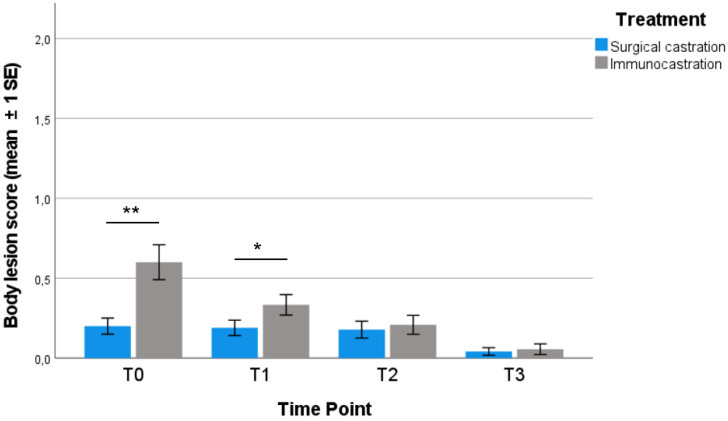
Mean (±1SE) of body lesion score in each TP for each treatment group considered (Mann–Whitney U test; * *p* < 0.05; ** *p* < 0.01).

**Figure 3 vetsci-09-00410-f003:**
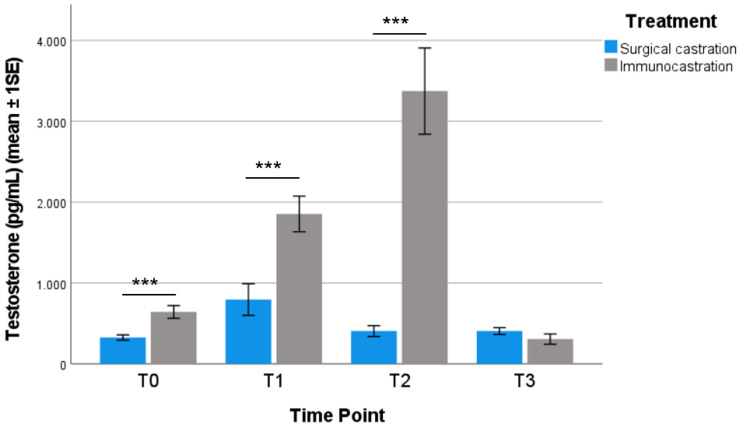
Testosterone levels (mean ± 1SE) in each TP for each treatment group considered (Mann–Whitney U test; *** *p* < 0.001).

**Table 1 vetsci-09-00410-t001:** Timing of vaccination with Improvac^®^ of the IC group, according to the age of pigs.

Age	Treatment
15 weeks	V1
22 weeks	V2
24 weeks	V2.2
32 weeks	V3
36 weeks	V4

**Table 2 vetsci-09-00410-t002:** Ethogram of behavioural categories and related description.

Behavioural Category	Description
Inactivity in lying	Animal is lying inactive (sleeping or resting)
Activity	Animal is busy in one of the following: walking, exploratory behaviour, social and agonistic interactions, sniffing, biting, chewing, or exploring environmental enrichment

**Table 3 vetsci-09-00410-t003:** Age and treatment group size at the beginning of the growing and fattening phase and at slaughter.

	SC No. of Pigs	IC No. of Pigs	Age Weeks
Beginning of growing phase	94	94	13
Beginning of fattening phase	83	83	22
At slaughter	68	71	41 (SC) 40 (IC)

**Table 4 vetsci-09-00410-t004:** Mean and SE of carcass traits (hot carcass weight, fat thickness, muscle thickness, and percentage of lean meat content) of SC and IC pigs.

Carcass Traits	SC	IC	Mann–Whitney U Test
Hot carcass weight (kg)	145.10 ± 1.31	150.54	*p* = 0.002
Fat thickness (mm)	32.31 ± 4.72	30.38 ± 0.59	*p* = 0.034
Muscle thickness (mm)	58.47 ± 6.48	55.34 ± 8.94	*p* = 0.028
Lean meat content (%)	50.86	51.67	

## Data Availability

Not applicable.

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
