# Peer review of "Does Immunocastration Affect Behaviour and Body Lesions in Heavy Pigs?"

_vetsci, 2022, doi:10.3390/vetsci9080410_

Round 1

Reviewer 1 Report

The article presents a very important topic for the industry. Immunocastration has over time become a relevant topic in the debate for the improvement of animal welfare. 

The article is well organised and structured in the right way. 

The results are consistent with the assumptions made.

I only recommend minor revisions with the aim of enriching the initial literature with interesting results found in this work

Di Pasquale, J.; Vecchio, Y.; Martelli, G.; Sardi, L.; Adinolfi, F.; Nannoni, E. Health Risk Perception, Consumption Intention, and Willingness to Pay for Pig Products Obtained by Immunocastration. Animals 202010, 1548. https://doi.org/10.3390/ani10091548

And also to add this work, which although focused only on sheep and goats, offers important insights to enrich the authors' reasoning.  

Zeng, F.; Yi, D.; Teketay, W.; Jing, H.; Sohail, A.; Liu, G.; Jiang, X. Recent advances in immunocastration in sheep and goat and its animal welfare benefits: A review. J. Integr. Agric. 202222, 299–309 

Author Response

Dear Reviewer 1,

We would like to thank you for revising our manuscript and sending us your positive and useful comments, which we have thoroughly considered. With great pleasure we send you the updated version of the article, which is now entitled: “Does immunocastration affect behaviour and body lesions in heavy pigs?”. We truly hope the modifications we have made are adequate and in line with your remarks, but we also trust that thank to your observations our work is much improved. Our responses to your questions are printed in blue.

Thank you again for your work.

The article presents a very important topic for the industry. Immunocastration has over time become a relevant topic in the debate for the improvement of animal welfare. 
The article is well organised and structured in the right way. 
The results are consistent with the assumptions made.
I only recommend minor revisions with the aim of enriching the initial literature with interesting results found in this work

Di Pasquale, J.; Vecchio, Y.; Martelli, G.; Sardi, L.; Adinolfi, F.; Nannoni, E. Health Risk Perception, Consumption Intention, and Willingness to Pay for Pig Products Obtained by Immunocastration. Animals 202010, 1548. https://doi.org/10.3390/ani10091548

And also to add this work, which although focused only on sheep and goats, offers important insights to enrich the authors' reasoning.  

Zeng, F.; Yi, D.; Teketay, W.; Jing, H.; Sohail, A.; Liu, G.; Jiang, X. Recent advances in immunocastration in sheep and goat and its animal welfare benefits: A review. J. Integr. Agric. 202222, 299–309 

Thank you very much for your positive comments. We read and considered the interesting articles you have suggested, they were added as citations in both introduction and discussion (L50 and L403-414).

Reviewer 2 Report

The work presented for review brings new elements to the current state of knowledge concerning the use of immunocastration in pigs. The authors paid attention not only to the production effects of pigs subjected to immunacastration, but also to behavioral aspects - aggressiveness and body injuries.

The purpose of the work is clearly stated. The material used for the research is sufficient, the research methods have been selected appropriately. What were the authors' guidelines for giving pigs the first injection of Improvac at 15 weeks, and not as recommended by the manufacturer after 8 weeks of age?

The tables and figures are clear. The differences between the different groups have been marked correctly. The conclusions from the conducted research are clear and result from the obtained research results.

Discussing the results against the background of other authors is very detailed.

The publications cited by the authors of the article are well selected. For the most part, the authors refer to the latest knowledge published in renowned scientific journals.

However, the authors' writing should be corrected - item no. 20, because the full names were written, and they should be replaced with the initials of the names.

Author Response

Dear Reviewer 2,

We would like to thank you for revising our manuscript and sending us your positive and useful comments, which we have thoroughly considered. With great pleasure we send you the updated version of the article, which is now entitled: “Does immunocastration affect behaviour and body lesions in heavy pigs?”. We truly hope the modifications we have made are adequate and in line with your remarks, but we also trust that thank to your observations our work is much improved. Our responses to your questions are printed in blue.

Thank you again for your work.

The work presented for review brings new elements to the current state of knowledge concerning the use of immunocastration in pigs. The authors paid attention not only to the production effects of pigs subjected to immunacastration, but also to behavioral aspects - aggressiveness and body injuries.

Thank you very much for your positive comments, we are glad you have appreciated our work.

The purpose of the work is clearly stated. The material used for the research is sufficient, the research methods have been selected appropriately. What were the authors' guidelines for giving pigs the first injection of Improvac at 15 weeks, and not as recommended by the manufacturer after 8 weeks of age?

Our aim was to evaluate the impact of immunocastration on welfare indicators in heavy pig production, for this reason, the initial protocol (including the schedule of administrations) was specifically proposed by the local supplier. The supplier suggested the protocol based on their previous experience in the farm involved in our study and on the genetics of animals.

We are aware that in literature different timings for Improvac administration (also for the first injection) are proposed. The other study about immunocastration for Italian heavy pig production (Pinna et al, 2015) proposed interventions at 10–11, 26–27, and 36–37 weeks of age.

Considering the timing of our study and comparing it to Pinna’s protocol, the first injection (at 15 w.) was certainly later (although similar – i.e. 16 weeks – to the one used in other studies like the one of Andersson et al, 2012 and Di Martino et al, 2018, cited in the paper), but the effective dose, namely the V2.2 (which also coincide with the timing originally proposed for the second administration), was prior, at 24 weeks. We might say the effect of immunocastration was achieved earlier, however we must notice that the study of Pinna and colleagues was focused on productivity and meat quality, so no information was provided about animal behaviour and welfare.

The tables and figures are clear. The differences between the different groups have been marked correctly. The conclusions from the conducted research are clear and result from the obtained research results. Discussing the results against the background of other authors is very detailed.

The publications cited by the authors of the article are well selected. For the most part, the authors refer to the latest knowledge published in renowned scientific journals.

Thank you very much for your positive comments.

However, the authors' writing should be corrected - item no. 20, because the full names were written, and they should be replaced with the initials of the names.

Thank you very much for noticing, we have corrected this oversight.

Reviewer 3 Report

I have reviewed the manuscript entitle “Does immunocastration affect welfare in heavy pig production?”. The aim of the present study is clear and interesting. In general, the manuscript is well written but I have some suggestion regarding the methodology. In particular, to assess the welfare is necessary a multiple evaluations respect the behavior and physiological evaluations.

In the present work, the only evaluation is the body lesion score and the animal activity.  I think a more detailed evaluation of the pig behavior and some additional physiological evaluation, as for example Cortisol/DHEA levels are necessary to support the aim.

In addition, the results of testosterone and the additional vaccination timepoint suggest that there was some problems in the experimental design and that it is not fully explained in the discussion.

Some general suggestions:

-        Did the pig have the tails?  Add this info to consider for the lesion scoring.

-        Number of animals: the authors declare (table 3) that some animals died or not reach production standards. Please more clear. For example, animals for the saliva samples are less that the total. Did The animals excluded afflict this sample size?

-       References: check recent publications.

Author Response

Dear Reviewer 3,

We would like to thank you for revising our manuscript and sending us your positive and useful comments, which we have thoroughly considered. With great pleasure we send you the updated version of the article, which is now entitled: “Does immunocastration affect behaviour and body lesions in heavy pigs?”. We truly hope the modifications we have made are adequate and in line with your remarks, but we also trust that thank to your observations our work is much improved. Our responses to your questions are printed in blue.

Thank you again for your work.

I have reviewed the manuscript entitle “Does immunocastration affect welfare in heavy pig production?”. The aim of the present study is clear and interesting. In general, the manuscript is well written but I have some suggestion regarding the methodology. In particular, to assess the welfare is necessary a multiple evaluations respect the behavior and physiological evaluations.

In the present work, the only evaluation is the body lesion score and the animal activity. I think a more detailed evaluation of the pig behavior and some additional physiological evaluation, as for example Cortisol/DHEA levels are necessary to support the aim.

Thank you for your useful comment, as you pointed out we focused on behaviours that are linked with pig welfare and body lesions related to aggressions. We highlighted this in both the title and the aim of the manuscript (L1-3 and L75-77).

In addition, the results of testosterone and the additional vaccination timepoint suggest that there was some problems in the experimental design and that it is not fully explained in the discussion.

The immunocastration protocol was specifically proposed by the local supplier. The supplier defined the protocol based on their experience and genetics of animals raised in the farms involved in our study.

Our intent was not to plan Improvac administration on the base of testosterone concentration, but rather to evaluate testosterone level achieved through the different timepoints, just before the subsequent administration.

We have added information about the vaccination schedule and the experimental design in Materials and Methods (L115-118) and in the Discussion (L358-359).

Some general suggestions:

Did the pig have the tails? Add this info to consider for the lesion scoring.

Pigs were all tail-docked. We have also collected some data about tail biting according to Welfare Quality Protocol, but we found only 8 cases, equally divided in the two groups and in different timepoints. No possible statistical consideration about these animals could be done, so we had not mentioned it. After your observation we added this information both in Materials and Methods (L156-157) and in Results (L244-245)

Number of animals: the authors declare (table 3) that some animals died or not reach production standards. Please more clear. For example, animals for the saliva samples are less that the total. Did The animals excluded afflict this sample size?

To assure that all the animals within the experimental groups maintained a good health standard at the moment of the administration of the vaccine, pigs identified through the rearing process as “problematic”, from a health point of view or showing an inadequate body weight, were separated and managed according to their conditions. In particular, 11 immunocastrated pigs and 11 surgically castrated pigs were excluded from the study during the growing phase, and the same happened during the fattening phase. The excluded animals did not cause any numerical disproportion in the groups. At the time of loading 1 IC pigs and 4 SC were not sent to the slaughterhouse due to inadequate body condition. The outcome of the excluded pigs was not known, however it was not linked to vaccine administration.

Saliva samples were collected from randomly selected animals at each timepoint. Our study did not focus on the single subject’s curve of testosterone, but rather on monitoring the general trend of the experimental groups.

We clarified and added information about these aspects in the text: L88-93, L201-208.

-References: check recent publications.

Thank you for this useful your suggestion. We enriched the article with other recent articles in both introduction and discussion (L496, L499, L569, L572).

Reviewer 4 Report

The study aimed to evaluate the effect of immunocastration on animal welfare in heavy pig production. Behaviour, body lesions, salivary testosterone levels and productive traits were compared between immunocatrated and surgically castrated male pig. Manuscript is clearly written and technically sound. The methods are appropriate and adequately conducted. The results are high quality and their interpretation and disussion are sound.

Please consider changing the title.

Author Response

Dear Reviewer 4,

We would like to thank you for revising our manuscript and sending us your positive and useful comments, which we have thoroughly considered. With great pleasure we send you the updated version of the article, which is now entitled: “Does immunocastration affect behaviour and body lesions in heavy pigs?”. We truly hope the modifications we have made are adequate and in line with your remarks, but we also trust that thank to your observations our work is much improved. Our responses to your questions are printed in blue.

Thank you again for your work.

The study aimed to evaluate the effect of immunocastration on animal welfare in heavy pig production. Behaviour, body lesions, salivary testosterone levels and productive traits were compared between immunocatrated and surgically castrated male pig. Manuscript is clearly written and technically sound. The methods are appropriate and adequately conducted. The results are high quality and their interpretation and disussion are sound.

Please consider changing the title.

Thank you very much for your positive comments. According to your suggestion we changed the title in “Does immunocastration affect behaviour and body lesions in heavy pigs?”
